# Trajectory of Teacher Well-Being Research between 1973 and 2021: Review Evidence from 49 Years in Asia

**DOI:** 10.3390/ijerph191912342

**Published:** 2022-09-28

**Authors:** Dongqing Yu, Junjun Chen, Xinlin Li, Zi Yan

**Affiliations:** 1Faculty of Education, Northeast Normal University, 5268 Renmin Street, Changchun 130024, China; 2Department of Education Policy and Leadership, The Education University of Hong Kong, 10 Lo Ping Road, Tai Po, NT, Hong Kong, China; 3Department of Curriculum and Instruction, The Education University of Hong Kong, 10 Lo Ping Road, Tai Po, NT, Hong Kong, China

**Keywords:** teacher well-being, review, descriptive quantitative approach, developmental trend, Asia

## Abstract

This review portrays a dynamic developmental trend in the teacher well-being literature in Asia between 1973 and 2021 using a descriptive quantitative analysis approach. A search of the Scopus database identified 168 journal articles across 46 countries and regions in Asia. This number of publications indicated a substantial change in the knowledge corpus, particularly during the pandemic, although overall production was still relatively low. Further results revealed diversity but an imbalance of research location, research type, research methods, data collection techniques, and research foci. A functionalist perspective may suggest that the knowledge base on teacher well-being is at a beginning stage. Recommendations for future research are proposed including cross-region collaborations, more developed research foci, using mixed-method approaches, high-quality qualitative research designs, innovative qualitative techniques, and diverse qualitative data collection techniques.

## 1. Introduction

Teacher well-being (TWB) is essential not only for teachers themselves [1] and their schools [2], but also for the future society [3]. Although there are ongoing debates about the definitions of well-being, one early definition from the World Health Organization [4] seems to be widely cited: “a state of complete physical, mental and social well-being and not merely the absence of disease or infirmity” (p. 2). Taken from the positive psychology literature, Van Petegem et al. [5] defined TWB as “a positive emotional state that is the result of a harmony between the specific context factors on the one hand and the personal needs and expectations towards the school on the other hand” (p. 34). Studies in the field of TWB have been reviewed and the definition synthesized as “an individual sense of personal professional fulfilment, satisfaction, purposefulness and happiness, constructed in a collaborative process with colleagues and students” [6] (p. 102). Teachers have frequently experienced a wide range of changes, challenges, complexities, and adversities in their everyday lives. With increasing demands for academic accountability and academic emphasis, scholars and stakeholders have been keener to dig into what school teachers should give their priority to, such as improving teaching quality and student outcomes. However, teacher well-being is believed to have been sacrificed in the current priorities [7,8]. Therefore, promotion of TWB has become one key solution to maintaining and enhancing quality education, as well as contributing to the sustainable development of teachers, students, schools, and even a society aiming to be a ‘wellness society’ which incorporates constant change and unprecedented uncertainties for a better world [3,9].

A good sign is that studies on TWB have blossomed in the past two years. However, so far, the main contributors in this field have originated from Western countries [7]. In the absence of a critical mass of literature from other contexts (e.g., Asia), educational professionals, academics, and policymakers may have inadequate understanding of how TWB has evolved and is applied outside of the mainstream contexts (e.g., West). As a result, in non-Western contexts, there is insufficient knowledge and ability for policy decisions and practice within teacher training and for assessment means to be adequately informed. Globalization, within the educational research sector which includes TWB, has created a tendency to ignore the social-cultural environment that, importantly, can “act as a mediator or filter to the spread of ideas and practices across the global resulting their adoption, adaptation or even rejection” [10] (p. 304). The often-dysfunctional results of policy and practice borrowed from the West and applied to the East have already been proved [11]. 

Many practitioners in Asian contexts have developed a concern for well-being knowledge about, and support for, the teaching profession as TWB has been considered to have been sacrificed for the sake of quality education [8,12]. Such a necessity has been stressed by academics who have concluded in their research that teachers lack adequate preparation in their lesson planning and are not sufficiently backed by their teacher and professional training programs to cope with the different well-being problems that they might encounter both personally as well as professionally [9,13]. It seems it is an appropriate time, therefore, for TWB’s knowledge base within the Asia context to be built upon and explored.

A genuinely worldwide knowledge about TWB would potentially inform and support teachers to cope with professional challenges which threaten their well-being, and help us understand how these affect individual teachers, classroom experiences, and schools, and identify possible solutions to support them. Perseverance, however, in the exploration of a research area through combining theoretical as well as empirical systematic examinations is necessary so that the gathering of relevant knowledge as well as discoveries in comprehension is enabled [14]. The tendency of employing stronger and distinct theoretical as well as methodological instruments in order for TWB to be developed has advanced over a period of time [15]. Conceptual instruments include explicitly elaborating upon as well as applying increased and varied theoretical frameworks within TWB research. Methodological innovations have been focused on using additional methodical methods in conducting studies. Through reviewing the various types of studies which are in existence within a certain field, this can be ultimately achieved [16].

There have been to date no TWB review studies conducted within Asian countries that have focused specifically on the ways in which trends have developed in research within this field. This reported review study, therefore, had the objective of seeking to reflect the designs of construction of knowledge in TWB and the topics that have been covered within the studies, the kinds of studies and methodologies employed especially within the Asian context from 1973–2021, and the ways in which researchers and scholars have developed and worked together over the last 49 years. Such an approach has permitted the designs of scholarly advancement within the Asian context leading to the widespread and historical expansion of the TWB field to be compared. This study’s findings will result in approaches for hastening development regarding TWB research and developing capacity, in addition to producing new knowledge [14]. The research questions prepared and delivered in this explanatory review are as follows:To what degree has the volume of teacher well-being research progressed and how has it progressed in terms of publication level, research localities, and research matters between the years of 1973 and 2021 in Asia?To what extent has the distribution of research type (e.g., empirical, theoretical, and review) of teacher well-being research evolved, and in which ways have they evolved in the period 1973 to 2021 in Asia?To what extent have the research methods (e.g., qualitative methods, quantitative methods, and mixed methods) that academics employ in TWB research advanced from 1973–2021 in countries located in Asia?

## 2. Relevance of Investigating Teacher Well-Being in the Asian Context

Five aspects are of relevance in investigating TWB in the Asian context. The first relevance is that the construct of TWB is context dependent. McCallum and Price [17] argued that well-being “is something we all aim for, underpinned by positive notions, yet is unique to each of us” as it varies with “individual, family and community beliefs, values, experiences, culture, opportunities and contexts across time and change” (p. 17). Hence, teachers from Asia may hold a different understanding of their well-being compared with their counterparts from the Western contexts. As the research on TWB has been swamped by Western studies, investigations are required from Asian countries that feature differing political, social cultural, and organizational structures. These efforts can produce connections between distinct sorts of TWB as well as contexts.

The second relevance concerns the impact of TWB on teachers themselves, their schools, and society. TWB has been shown to influence individual teachers, particularly their professional values [18], meaning of work [19], their social relations [18], work contentment [20], lifelong learning [21], burnout [22], work commitment [23], and job performance [24]. Moreover, TWB has played a vital role in flourishing schools in the form of teaching improvement, student outcomes, school effectiveness, and educational governance [9,13,25,26]. From a systemic perspective, teachers influence their students, which then affects school well-being and wider communities [3].

The third relevance is the alarming current situation of TWB. Recent review studies on TWB have revealed that teacher occupational well-being is at risk worldwide due to the high burden of the teaching profession, increasing academic accountability, challenging societal demands on teachers, and complex management mechanisms [7,27,28]. Teachers in Asia are not an exception and, possibly, the state of TWB there since the pandemic has become even worse. Choy [29] reported that teachers from Hong Kong rated their level of work-related stress at 6.97 out of 10 during the coronavirus pandemic, up from a level of 6.4 before the health crisis. Some 35% of those surveyed worked more than 60 h a week, while nearly 15% worked 71 h or more. This alarming situation has been echoed in the Philippines [30]. These statistics are appalling and urge us to focus on the pressing need to support teaching professionals.

The fourth relevance concerns the unique teacher policy mechanisms in Asia which may have an impact on TWB. Teacher instruction and assessment in the majority of Asian countries have been propelled by expanding academic responsibility. This has been complemented by critical exams which appear to have downplayed feelings and exclusively prioritized success [31]. More recently, policy discussions have arisen with regard to linking teacher salaries with assessments of ‘merit’, whereby teachers are appraised and given appropriate promotions [6]. Such a focal point has had an influence on classroom procedures in many ways because of the kinds of exchanges that can occur between teachers and students, the behaviors of students [32], and how well they perform in school [33]. Studies have indicated that an increased prominence of critical testing and answerability is shifting the essence of classroom communications [34], which is linked with the heightened numbers of teachers who are leaving the profession [35] and encountering stress [36], pressures, and high levels of anxiety [37]. Accordingly, TWB has been unavoidably affected. 

The fifth relevance is that, fortunately, TWB literacy is malleable and trainable. Well-being literacy has been defined as “the capability (incorporating knowledge, vocabulary, language skills) of comprehending and composing well-being languages, sensitive to contexts, used intentionally to maintain or improve the well-being of oneself or others” [38] (p. 327). Well-being literacy centers on capacities that will strongly maintain positive functioning. It is an individual’s mechanisms for change inside. Altogether, TWB is conceptually, empirically, and practically significant and, consequently, it is worth further investigation in order to unveil the true nature of TWB literature and the trajectory of its knowledge accumulation in the Asian context. 

## 3. Method

To ensure a systematic review of the qualified articles, detailed and systematic steps were implemented, guided by the Preferred Reporting Items for Systematic Review and Meta-Analysis (PRISMA) [39]. The review had four phases, namely establishing the inclusion and exclusion criteria, literature search and study identification, data screen and extraction, and data analysis.

### 3.1. Inclusion and Exclusion Criteria 

To make certain that the articles found were of relevance to the specified aims, criteria for ascertaining their inclusion were formulated preceding orderly searches being conducted. First, journal articles needed to be published between 1973 and 2021, and data locations had to be in Asia. Articles first published online in 2021 were also included. The rationale for choosing this time period was that the literature of TWB first emerged in 1973 based on our initial search [40]. Second, the research topic had to be relevant to ‘teacher well-being’ or at least dimensions/elements of TWB. Third, research participants had to be, or include teachers, including pre-service and in-service teachers. Fourth, journal articles had to be peer-reviewed to ensure their quality. Fifth, research papers had to be written in English. Sixth, it was a requirement that the research studies were empirical in their design, review style papers, conceptual, or commentary type documents. 

Furthermore, the following exclusion criteria were established. First, papers where ‘teacher well-being’ was not specified as, or at least part of, the study purpose. Second, papers where TWB only played a marginal role. Third, papers where teachers were not listed as the participants. Fourth, papers whose main text was not written in English even though the titles, abstracts, and keywords were in English. Fifth, other types of research publications such as book chapters, books, letters, notes, editorials, conference papers, and conference reviews.

### 3.2. Literature Search and Study Identification 

The searching process began with a systematic search for abstracts of potentially relevant studies in early 2022 in SCOPUS with keywords of educator* OR teacher* OR teaching* and well-being* OR wellbeing*. To avoid the potential risk of publication bias and the omission of qualified papers, the researchers employed two recommended strategies. First, researchers searched the identical keywords in Google Scholar for peer-referred articles published in the same time period to find any missing articles and to cover grey literature [41]. Next, a “snowballed” strategy was utilized by searching through the reference lists of the widely cited [6,27,42] and the most recent reviews [7,43]. As a result, 168 articles formed the initial corpus.

### 3.3. Data Screen and Extraction

The Preferred Reporting Items for Systematic Review and Meta-Analysis (PRISMA) guidelines were strictly obeyed in this reported review [39,41]. To measure the appropriateness of the saved articles subsequent to their passing the specified criteria used for including or excluding them, the following four steps were adopted to screen and select the articles. These steps were *identification*, *screening*, *eligibility*, and *inclusion*. The ***first*** step was *identification*. Through the SCOPUS filter first, and Google Scholar as the supplementary data base to cover grey literature, 10,466 articles were identified by scanning the titles, keywords, and abstracts. Specific considerations were given to the topic appropriateness, book chapters or commentaries, and whether the complete documents had been written in English. In certain cases when the abstract supplied inadequate data to accurately verify its eligibility, the complete document was subsequently checked by one of the researchers (namely, the first researcher). The *second* step was *screening*. One researcher read all abstracts to ensure the eligibility of the articles in terms of topic relevance, educational level, participants, research type, and language. For example, papers which happened to get through the SCOPUS filter, but were studying how teachers could improve students’ well-being, were excluded at this step. As the searching result was generated in one database, the risk of duplication was avoided. When the abstracts provided too little information, full articles were downloaded and scanned by one researcher for further decision. If questions were raised as to whether an article should be included or not included in the review, the complete article was then acquired and autonomously vetted by the first researcher. During this procedure, 10,280 papers were screened out and 186 were retained. The third step was eligibility. At this step, full articles were downloaded and read by the same researcher for more detailed examination. Particular attention was given to whether TWB was placed in a central or marginal role, and also to article quality issues such as written too briefly, only vague information about research aims given, samples (if any), research methods, results, and/or other major contents. Any doubts or questions during this procedure were discussed with the second researcher to reach an agreement. During this step, 10,094 articles were screened out and 186 articles were retained. The fourth step was *inclusion*. The second researcher checked the whole set of the data base to ensure the accuracy of previous procedures and selection of the articles. The third researcher joined for discussion if there was disagreement between the first and second researchers. As a result, 18 papers were screened out and a total of 168 articles was reserved for analysis (see Figure 1 for the flow chart). 

### 3.4. Data Analysis

A descriptively quantitative analysis approach is considered appropriate for reporting the trajectory of a certain topic in a given time period [44]. In this study, this approach was utilized to portray the developmental patterns and variations in TWB research in Asia over the past 49 years. Publications that adhered to the inclusion principles were initially coded by employing a self-created data instrument for extracting data in the Microsoft Excel program that was designed for the researchers [14]. The instrument that was employed for extracting the data comprised of the date on which the article was published, the data setting, the topic foci, the kind of research, and the research methodology. Methodological styles were a central focus in order for matters regarding methodologies to be appropriately addressed. The instrument used for extracting the data that was accessible to the research team was pilot checked on five of the chosen articles by two of the researchers and adjusted according to the coding form.

## 4. Findings

The findings of this review were acquired from the 168 TWB articles with research conducted in Asia and published between 1973–2021 by assessing the year of publication, the data location, the foci of the research, and the type of research, in addition to the methodology employed. From these, 167 (99.40%) empirical publications, 1 (0.60%) conceptual/commentary publication, and zero review articles were found. It needs to be kept in mind that empirical publications were incorporated within the calculations that were made regarding the location used for collecting the data and the research methodology employed.

### 4.1. Year of Publication

Generally, the analysis of the differences with regard to the volume of journal publications revealed an expansion over a certain period of time. This information is displayed in Figure 2. The 49-year review span was distributed into ten equal 5-year time periods but not, however, for the last period of four years. The articles published in the final time period between 2018 and 2021 were 118 (70.24%) in number, which encompassed the largest segment of the total corpus. This was sequentially and decreasingly followed by the other intervals going back in time. There was a remarkably rapid growth from the eighth interval (2008–2012) to the ninth interval (2013–2017), as the publications increased by more than triple from 9 (5.36%) to 34 (20.24%). Likewise, the number in the last interval reached a peak that was more than three times the previous period, from 34 (20.24%) to 118 (70.24%). The research team reached the conclusion that during the first seven 5-year time periods, few studies about TWB were conducted in Asian places, while the amount has soared dramatically to high numbers during the past 14 years. 

### 4.2. Data Location

Locations from which data were collected were available from 167 identified empirical publications out of a total number of 168. These locations were found to be 46 countries and areas. One conceptual/commentary paper did not indicate the location where the data were gathered. The top ten data locations were Mainland China (36, 21.56% out of 167 empirical articles), followed by Hong Kong (25, 14.97%), India (18, 10.78%), Israel (17, 10.18%), multiple countries (16, 9.58%), Turkey (14, 8.38%), Malaysia (12, 7.19%), Iran (9, 5.39%), Pakistan (7, 4.19%), and the Philippines (7, 4.19%). These 10 regions accounted for the majority of the published research (145, 86.83%). The residual research production was contributed by the remaining 36 regions in Asia. For instance, Indonesia and South Korea each produced five articles (2.99%), Singapore and the United Arab Emirates each produced three articles (1.80%), and Oman, Argentina, Japan, Palestine, Saudi Arabia, Taiwan, and Macao each had two articles (1.20%) 

Table 1 illustrates the observed trend found within the first ten data location areas within the ten time periods between 1973–2021. The first article regarding TWB in Asia was produced with samples from India. After that, only five studies (2.99%) could be located for the following six periods (1978–2007). Subsequently, steady expansion could be witnessed with regard to the number of publications that were generated within each area in the succeeding time periods. The proportion in Mainland China remained the largest in its total number, far greater than for other countries. However, from a horizontal view, the publication volume in Mainland China only became dominant in the last interval. It is interesting to note that there was only one study from there (0.60%) in both the eighth and ninth periods, and then the number exploded in the last four years (2018–2021), jumping to 34 articles (20.36%).

### 4.3. Research Foci 

Formulated using the foremost concepts found in the identified publications, the research team coded the research foci into two distinct and agreed-in-advance levels. Three key research themes were identified at the initial level. These were: TWB’s nature, which equated to 68 (40.48%) out of the total 168 articles; antecedents and/or the correlates of TWB, which amounted to 140 (83.33%); and finally, TWB consequences, which were calculated to be 41 (24.40%) of the located publications. These are displayed in Table 2. The three major themes or topics were then divided into various secondary themes to exhibit the research foci. The first key research theme, namely TWB’s nature, is comprised of five second-level groupings. The first of the sub-themes is TWB’s theories, which equated to 35 (51.47%) from the total 68 articles that had a focus on TWB’s nature; for instance, Self-Determination Theory by Meng [45], Broaden and Build Theory by Greenier et al. [46], and Job-demands theory by Li et al. [47]. The second sub-theme is definitions of well-being and TWB (35, 51.47%); for instance, subjective well-being by Kurt et al. [48] and mental well-being by Sun et al. [49]. The third sub-theme is components of TWB (23, 33.82%); for instance, stress and work engagement by Alqarni [50], job satisfaction by Song et al. [51], and life satisfaction by Ngui and Lay [52]. The fourth sub-theme is models of TWB (3, 4.41%); for instance, Effort-recovery Model by Gu et al. [53] and Cognitive-Development Model by Balakrishnan et al. [54]. The fifth sub-theme is measures of TWB (5, 7.35%); for instance, validation of Utrecht Work Engagement Scale by Klassen et al. [55] and Multiple Abilities and Subjective Well-being of Kindergarten Teachers by Wei [56]. The second first-level theme, antecedents of TWB, consists of three sub-themes, namely, contextual antecedents (5, 3.57% out of 140 articles targeting antecedents) (e.g., integration of technology [57]); organizational antecedents (68, 48.57%) (e.g., relatedness [58]); and individual antecedents (106, 75.71%) (e.g., depression [59]). The third first-level theme, consequences of TWB, includes four sub-themes. The first sub-theme is teacher personal outcome (32, 78.05%); for instance, teacher task performance by Cho [60]. The second sub-theme is teaching practice (6, 14.63%); for example, teachers’ innovative teaching by Rafsanjani et al. [61]. The third sub-theme is student learning and outcomes (4, 9.76%); for instance, students’ academic performance by Zach and Inglis [62] and emotional and disciplinary relationships by Asl Marz et al. [63]. The fourth sub-theme is school-level outcomes (2, 4.88%); for instance, school turnover rate by Karakus et al. [64].

Notably, as shown by Figure 3, there were almost no studies concerning the nature of TWB in Asia in the first seven periods (1973–2007), except for one study (1.47% out of 68) in the sixth period (1998–2002). Despite the huge disparities between the numbers and rates, more apparent positive trends can be found since the eighth period (2008–2012) in four sub-themes, namely theory, definition, content, and measure, while the developing pattern of studies focusing on models of TWB remained unstable until the present. 

Similar to the studies regarding TWB’s nature, obvious upward tendencies can be observed among articles about antecedents of TWB since the eighth period (2008–2012) with nine studies (6.43% out of 140). Since then, research output in this regard has grown at a surprising rate, with the volume of new papers tripling in the last period from the penultimate period. Numbers of research papers concerning all three themes of antecedents increased, among which the most effort was paid to studying individual factors (106, 75.71%), leading to the biggest share and fastest increase being in this branch. 

Interestingly, the period when research about the consequences of ECE TWB sped up was even later. Only in the last four years (2018–2021) did research regarding consequences first exceed double digits and reach a high level of 32 papers (78.05% out of 41), which is more than five times the number in the previous period (6, 14.63%). The influence of TWB on teachers’ personal outcomes and teaching practice also showed promising positive signs, while research focusing on students’ learning and outcomes and school-level consequences received much less attention and accumulated only four studies (9.76%) and two studies (4.88%), respectively.

### 4.4. Research Type

Articles were categorized into three types, namely, empirical articles, review articles, and conceptual/commentary articles (Figure 4). During the past 49 years covered in this review, 167 (99.40%) empirical research studies exemplified the bulk of TWB’s international knowledge situated within Asian localities, with one (0.60%) paper that was considered to be conceptual or commentary, and nil review type studies. Clearly, empirical studies have been the most preferred type of research for scholars investigating TWB within Asian locations. Although there were relatively few empirical studies in the first seven 5-year intervals, the number of articles in the eighth period, 9 (5.36%), was more than four times as large as in the seventh, 2 (1.19%), implying an upward trend. This number consecutively tripled in the last ten years, rising from nine articles (5.36%) to 34 articles (20.24%) (2013–2017), and finally to 117 articles (69.64%) (2018–2021). Quite the opposite, just one (0.60%) conceptual/commentary paper has been published and this only occurred in the last 5-year interval between 2018 and 2021.

### 4.5. Research Method

Separate follow-up analyses for each of the methods were formulated. Initially, the research team grouped empirical publications corresponding to whether they employed a qualitative method, a quantitative method, or mixed methods. Next, we micro focused, identifying five distinct levels of numerical methods which were employed within the quantitative or mixed methods investigations. Finally, the research team assessed the sorts of qualitative data gathering techniques applied within the investigations that used qualitative and mixed methods. 

First of all, the empirical publications were classified as to whether they had used qualitative, quantitative, or mixed methodologies. It was found that, overall, academics who were researching TWB in the Asian context expressed a clear inclination for using quantitative approaches. Amongst the 167 located empirical research papers, 147 (88.02%) of them employed a quantitative research model, with 12 (7.19%) of them adopting a mixed method model, and 8 (4.79%) employing qualitative methodologies. Figure 5 depicts the developmental trends of these three research methods over the 5-year intervals during the chosen 49 years. In the first seven periods (1973–2007), the numbers of quantitative studies were small and fluctuating. The first apparent rise appeared with nine studies (5.39%) in the eighth interval (2008–2012) growing to 29 (17.37%) in the ninth interval (2013–2017). This number then almost tripled in the last five years, growing to 103 (61.68%). Articles that were constructed by using a qualitative model commenced being seen within the seventh period, which was between the years of 2003 and 2007, and underwent constant progress within the last two periods of time, following an intermission with zero output during 2008 to 2012 (the eighth period). Despite the small quantity, the number of such articles reached a peak in the last five years (between 2018–2021) with five (2.99%). It was not until the years 2013–2017, or the ninth interval, when mixed method designs were apparent in exploring TWB within Asia. Subsequently, there was a significant increase within the last interval, growing from three (1.80%) research investigations to nine (5.39%) publications in the final time period, although it remained a modest percentage of the entire corpus. It should be remembered that 151 of the studies (90.42%) opted to use first-person statistics (e.g., teachers’ reactions), 16 of them (9.58%) applied an experimental research model, and finally, 13 (7.78%) were classified as being intervention type studies.

Secondly, we narrowly focused on the subgroup of quantitative investigations in order to identify the statistical level that every investigation had attained. It should be noted that each article was coded at one level only, such as the highest level the data analysis had engaged in. In order for this analysis to be facilitated, the located data were coded. There were five distinct statistical method levels, which followed an adapted form of the scheme which was used by the research team for classification purposes [65] (please refer to Table 3).

Figure 6 reveals a continual increase regarding the studies which were located at each level of the statistical analysis within each of the 5-year intervals. Astonishingly, among the 159, or 95.21%, of the empirical studies which made use of quantitative methods, more than half of them opted to employ Level 5 statistics (87, 54.72% from 159 articles), with Level 4 statistics (34, 21.38%), Level 3 statistics (17, 10.69%), Level 2 statistics (15, 9.43%), and Level 1 statistics (6, 3.77%) following thereafter. It is noted that the study using Level 1 statistics appeared the earliest, in the first period (1, 0.63%). However, there was no research output using only Level 1 analysis in the following seven 5-year periods. Coincidentally, during the period of 1983 to 1987 (5-year interval), one study was located that employed Level 5 statistics. There was no research using Level 5 statistics in the following four 5-year periods, but the number grew vigorously to 13 (8.18%) in the ninth period and 68 (42.77%) in the tenth. Similar trends were shared by the other three levels, but with slower paces and less impact compared with Level 5. It should be pointed out that the cross-sectional model was, and continues to be, a feature of research investigations utilizing quantitative methodologies. Amongst the 159 quantitative or mixed method publications, 147 (92.45%) of them opted to utilize a cross-sectional methodological design, whereas 12 of them (7.55%) utilized a longitudinal design. 

Next, the research team inspected the types of qualitative data gathering techniques which were used for the 8 qualitative (4.79%) and 12 mixed method publications (17.19%). The usage of a variety of qualitative data strategies for gathering data by five-year periods of time is displayed in Table 4. In general, studies favored mostly using interviews, qualitative questionnaires, and classroom setting observations for collecting their data, while more wide-ranging techniques were, during the past ten years, also employed in qualitative investigations. There was no qualitative research in the first six 5-year intervals (1973–2002). In the seventh and eighth time periods, one single article (namely 5% from 20) chose to apply a qualitative method in order to collect its data, by using interviews as its preferred methodology. In the last nine years (2013–2021), more diversity is evident in qualitative methods, which include interviews (15, 75%), qualitative surveys (2, 10%), classroom observations (2, 10%), case studies (1, 5%), reflective reports (1, 5%), and journal writing (1, 5%). The distribution was highly skewed by researchers’ preference for interviews (16, 80%). Moreover, it seems that a positive trend is only recognizable for this technique, while the development patterns of the other techniques appeared much flatter.

## 5. Discussion and Future Research

This section will outline, interpret, and discuss the major findings using publication year, data location, research foci, research type, and research method from the 168 articles on TWB in Asia published between 1973 and 2021. Recommendations for future research will be presented subsequently. 

### 5.1. Publication Year

Analysis of change in the quantity of the journal articles on TWB showed a continuous increase over 49 years and also showed it has increased remarkably over the two final intervals starting from 2013. The 168 articles that were conducted in the Asian context, despite not an impressive number, had attained the critical amount required for this study’s symmetric review to be conducted [14]. By the eighth interval (2008–2012), still only a handful of articles (9.52%) on teacher well-being were published in Asia. The number in the last interval reached a peak, and it was more than three times that of the previous period. One influential systematic review of the global literature on teacher well-being was conducted by Bricheno et al. [66] who called for more attention to teacher well-being. The developmental styles of TWB research could have been possibly encouraged by this conducted review. Within the last time period, the number of publications amounted to 118, contributing to the biggest percentage of the overall research corpus. It is worth pointing out that the number of articles (77) strongly increased since the pandemic happened in 2020 and 2021. This trend is representative of a significant change which has occurred within this research field. Projecting ahead, the trends documented in this review portend a far more balanced intellectual framework for teacher well-being research in the future, though we are not sure whether this boosting trend is a temporary phenomenon due to the unexpected pandemic, or whether scholars have now become aware of its important role in education [14].

### 5.2. Data Location

Data location also generated a fascinating trend. Around half of the Asian-focused teacher well-being research (75, 44.64%) has been conducted in the top three locations, namely, Mainland China, Hong Kong, and India. Although not the earliest publication country, the leading region, Mainland China, stands up in the final period as producing the largest portion of the total knowledge corpus, far more than those in other societies. Looking more closely, approximately 14.97% of those were published during 2020 and 2021 during the pandemic. This trend is shared by other data locations, which is consistent with the evidence on the volume observed during the pandemic. Hong Kong (14.97%) closely followed and this led to producing more teacher well-being articles versus three (2008~2012) and two (2013~2017) intervals ago. The standing of India also demonstrated that the increase in the teacher well-being field largely attaches to the contextual environment of the pandemic. Scholars from Mainland China [67,68] and Hong Kong [69,70] have put much effort into well-being research in education, which may result in their continued leading status in Asia. Additionally, considerable variations were observed between the various regions. The literature on TWB can be seen as being greatly skewed due to the research contributions originating mainly from a small number of academically active places (that is, Mainland China, Hong Kong, India, Israel, and Turkey) that share similarities but also large differences. The variation was also apparent between societies that can be classed as not being as academically active (for example, Oman, Argentina, Japan, Palestine, and Saudi Arabia), and many other Asian regions kept salience. Such trends restrict any portrayal of an intellectual worldwide structure due to our knowledge base remaining highly inhibited by its potential geographical reach [14]. The precise explanations causing this phenomenon are unknown. One feasible way to meet the task of variation between societies in the near future is for shared research capacity to be drawn upon by academics residing in varying societies, countries, and regions. The good news is that this review affirms a positive signal for international collaborations in the teacher well-being field, that is, 9.58% (16 articles) covered more than one data location. This observation implies that research capacity exists in a certain number of societies which have tried to leverage that capacity with those in other contrasting societies through cooperative ventures to advance teacher well-being research [16]. 

### 5.3. Research Foci

It is interesting to note that the ‘Asian knowledge base’ showed a skewed distribution by focusing on limit variations of the foci, namely, the antecedents and/or correlates, nature, and consequences of TWB by a descending sequence. Data showed that research on TWB focused mainly on influential factors, particularly individual [59,68] and organizational drivers [50,58], over time. This observation may indicate that the scientific world has gradually realized that TWB is threatened and has put efforts into investigating its influential drivers in order to promote TWB. This result is echoed by a recent review [7]. However, that comprehensive review did not reveal the patterns and trends over time of TWB literature. In our review, although it was the least prevalent research focus, investigation of the consequences of TWB seems to have increased in the final interval [67,71]. Out of those studies of the effects of TWB, teacher personal outcomes [60] ranked the first, followed by teaching outcomes [61], student learning and outcomes [62], and school-level outcomes [64]. The scarcity of investigations with this research focus may be because the current research often regards TWB itself as an outcome variable [57,72]. It is also of significance, however, that very rarely has the impact of TWB on other stakeholders and the wider environment been explored. By examining the theme of the nature of TWB, this review observed sub-themes including theories [45], conceptualization [48], components [50], and models of TWB [53]. Although scholars have paid attention to the nature of TWB, it is still claimed that mis-conceptualization has existed and agreement on the definitions has not yet been reached in the TWB literature [7]. Furthermore, investigations on integrated models and measurements have also been skimpy in the past 49 years.

The research themes discovered in this review look to be convergent in addition to being divergent. When considering the convergent standpoint, the studies assessed had a tendency to concentrate on three foremost themes: the antecedents, the nature, as well as the effects of TWB. These have provided suggestions for how the intellectual composition of TWB can be better understood. From the divergent perspective, the TWB field contains diverse topics, such as the antecedents and/or correlates, nature, and consequences of TWB and their sub-themes mentioned above. This observation highlights not only the importance of prioritizing the research agenda of teacher well-being, but also the desirability of broad and balanced coverage in the selection of research topics. This finding points to the need for programmatic research for understudied topics, such as the effects of teacher well-being beyond teacher personal outcomes, and the authentic nature of teacher well-being. This finding may also promote clarifying the fragmented conceptualization of teacher well-being [73] and add value to the significance of teacher well-being for organizational outcomes [7]. Furthermore, TWB’s theoretical point of view should attract more notice as theoretical as well as conceptual frameworks within the TWB field are somewhat lacking in number [1]. Advancement in tackling such vital dilemmas necessitates a sustained effort on these dilemmas and research gaps, which should also be repeated by the work of other academics [27]. 

### 5.4. Research Type

As expected, empirical articles not only represented the largest portion of the literature, but also accounted for almost all of the overall volume of publications, since only one commentary article has been produced and no review articles of the Asian literature have emerged during the past 49 years. This distribution seems largely out of balance. Aligning with the developmental trend, empirical publications substantially increased during the last three intervals starting from 2013. The only commentary article [74] appeared recently. The Asian literature in the current study has identical features when compared with other reviews of TWB [7,27,75]. When considering the level of progression of studies on a worldwide scale, concentrating on empirical research output would seem to be the most suitable and relevant. It has been claimed in the literature [76] that a sound knowledge base needs to be built upon a sizeable collection of higher quality empirical investigations. However, there is also a strong need for high-quality systematic reviews and conceptual efforts on teacher well-being which will strengthen the quality of future research. 

### 5.5. Research Method

Not surprisingly, scholars have demonstrated a notable preference for quantitative research methods over time, with only a limited number of qualitative and mixed methods in the TWB literature in Asia. This sole preference stands until the seventh period with the emergence of qualitative design and the ninth interval with mixed method design. A detailed examination found that a larger than expected percentage of scholars were employing advanced statistical methods, despite a steady rise in the number of studies at each level of statistical analysis in each interval. This trend seems more consistent in the final two intervals. Such reflections present a general overview of the research method patterns found within TWB studies which have been conducted in Asian localities. The reassuring feedback is that the employment of different procedures (quantitative, qualitative, and mixed) has a tendency to become more disparate over a period of time and more sophisticated quantitative practices have become apparent in research, notably within the last interval. However, these trends also represent an imbalanced development regarding research methods, as qualitative and mixed method designs largely made less development progress in the analysis.

Moreover, from the 159 studies utilizing quantitative methods, 147 of them were cross-sectional design (92.45%) [49,68]. Such an observation requires extra testing due to the cross-sectional research design only providing a temporary report. A total of 151 studies employed first-person statistics (e.g., teachers’ reports); 9.58% of them applied a design which was classed as being experimental research; and 7.78% of them were classed as being intervention studies. Taken together, the strong presence of first-person, cross-sectional, and quantitative methods in the teacher well-being literature in the past may be a bias caused by the current investigation having focused more on understanding the state of teacher well-being and its influential drivers. Such findings enhance our picture further with regard to the Asian research context being largely undeveloped; however, it also is promising with regard to future capability and research excellence in the methods used to conduct research investigations [76].

Additionally, data revealed that more diversity has been apparent within qualitative collection practices beginning from 2013. The distribution of collection techniques was highly skewed by researchers’ preference for interviews and the trend appeared to continue. It was also observed there was a lack of innovative qualitative techniques. Using more innovative qualitative techniques to measure TWB has also been suggested by other scholars [7,75].

To increase knowledge production, an overall balanced development is generally considered desirable. In addition, there is an expectation that qualitative research designs exist in the opening stage and develop upon strong groundwork for interpretation and conceptual production [76]. This would appear to not be what is happening as far as TWB’s production of knowledge within Asian countries is concerned, as shown by the quantitative research designs and when they were initially used. Whatever the reasons for the research design, a diversity and balance of research location, research type, research methods, data collection techniques, and research foci is desirable in order to contribute to the development of knowledge production on TWB. It seems that the knowledge base on TWB has achieved diversity for each aspect to a certain extent, but an imbalance has been also identified. From this perspective, the knowledge base in TWB in Asia seems to be at the beginning stage as evaluated by the functionalist approach [76].

Based on the evidence above, several recommendations are proposed. First, cross-region collaborations are encouraged, which may scale up the balance of the overall volume and data location. Second, more efforts should be directed towards underdeveloped research foci, such as the integrated TWB model and robust measures and consequences of TWB (i.e., teachers, teaching, learning, and school level outcomes). Third, in future investigations, applying a mixed-method methodology ought to be urged, so that findings are more varied and credible [77]. Fourth, although research methods seem to be balanced in terms of diversification, more studies using longitudinal design and reciprocal modelling design should be included in the next research agenda to bring teacher well-being research to maturity [76]. Fifth, there is a need for additional qualitative research designs of higher quality to be thought about. As a result, these will increase the number of investigations looking at the nature as well as the effects of TWB, and the formation of a conceptual model. These types of topics are currently underdeveloped. Sixth, there is a lack of innovative qualitative techniques to authentically assess teacher well-being in addition to self-reported methods. Seventh, and finally, the most commonly applied techniques used for collecting qualitative research data are primarily reflective in their nature (e.g., interviewing, accounts of people’s life experiences, group conversations), which may not be satisfactorily precise as they are dependent upon the exactness and lucidity of human memorization [78]. Therefore, strategies, including observations and machine monitoring devices, need to be encouraged for collecting first-hand accounts and instantaneous well-being knowledge in genuine circumstances.

## 6. Limitations

In this review, two major limitations are now highlighted. One limitation was that we focused on the trajectory of teacher well-being literature rather than on the specific content of the reviewed publications. The research team decided not to specify any understandings of the reviewed studies. In a separate qualitative systematic review conducted by the current research team [79], in-depth examinations of the teacher well-being literature were made. Secondly, a mindful choice was made that our survey of publications conducted on teacher emotions would be restricted to journal articles which went through the peer-review process. It could be contended that a greater selection of accessible findings, perhaps located in books, chapters in books, study reports, and theses/dissertations, was not covered. The findings of our review, therefore, may have been different from what would have been reported if the aforementioned had been included.

In conclusion, the current descriptive review portrays a dynamic trajectory of teacher well-being in Asia which fills in the gaps regarding the overall developmental trend in the past half century. Imbalances regarding the research foci, data collection locations, research types, research methodologies, and data collection procedures, however, were also detected. Therefore, future research may pay more attention to these points in order to contribute to advancing the knowledge production of TWB, which will also assist policymakers and practitioners in constructing a fuller understanding of the TWB concept and knowledge base, not only so that teachers can better handle the constant change and unprecedented uncertainties in the future, but also so that teacher well-being will not be sacrificed for the sake of quality education [8].

## Figures and Tables

**Figure 1 ijerph-19-12342-f001:**
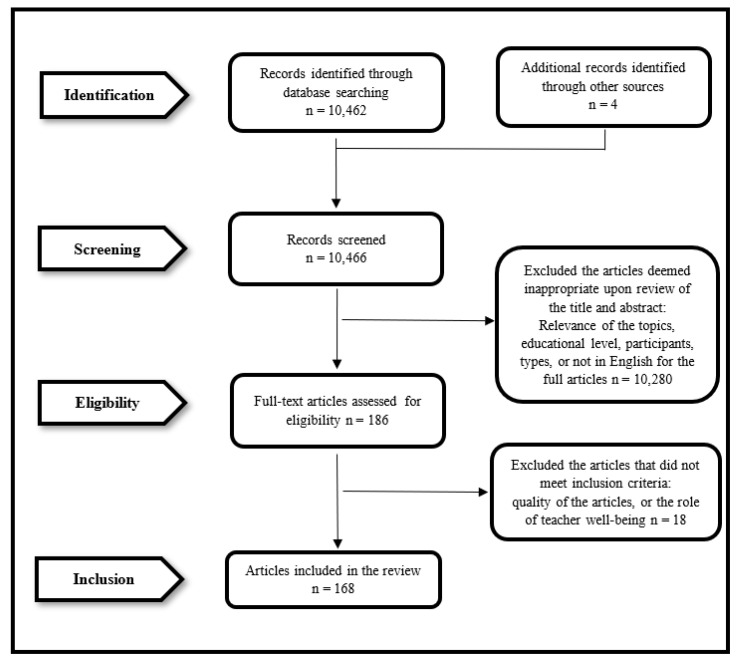
Flow Chart of Search and Screening Process.

**Figure 2 ijerph-19-12342-f002:**
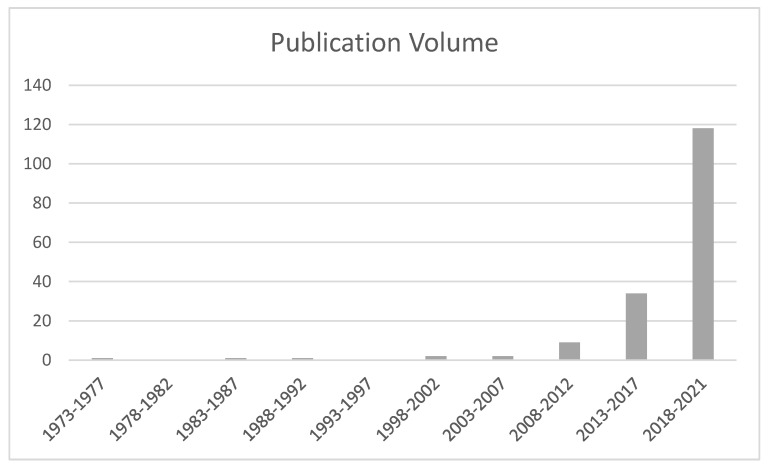
Trend of Publication Volume by Five-Year Interval during 1973–2021 (N = 168).

**Figure 3 ijerph-19-12342-f003:**
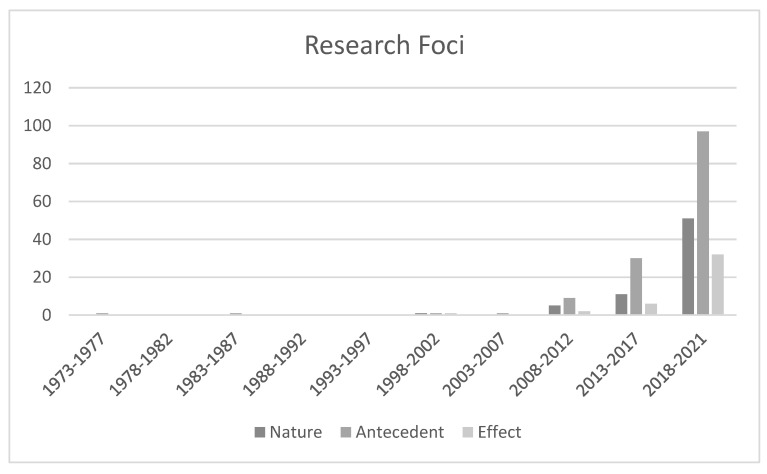
Trend of Research Foci by Five-Year Interval during 1973–2021 (n = 168).

**Figure 4 ijerph-19-12342-f004:**
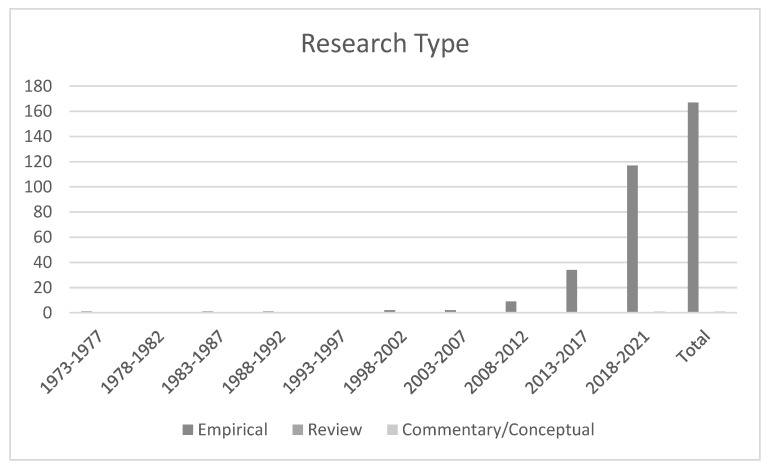
Trend of Research Type by Five-Year Interval during 1973–2021 (n = 168).

**Figure 5 ijerph-19-12342-f005:**
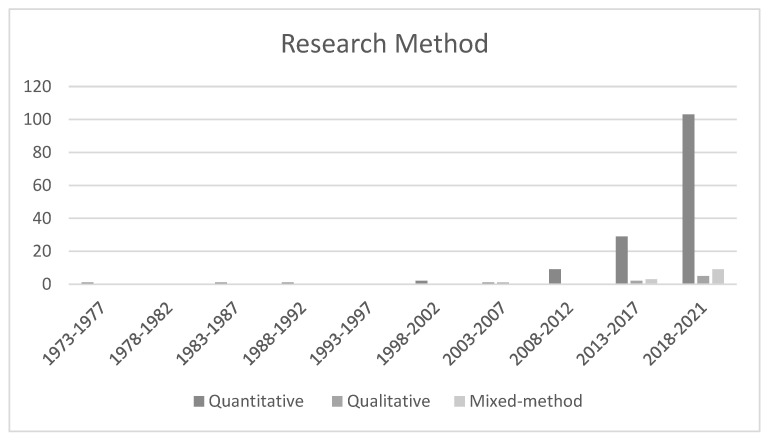
Trend of Research Method by Five-Year Interval during 1973–2021 (n = 167).

**Figure 6 ijerph-19-12342-f006:**
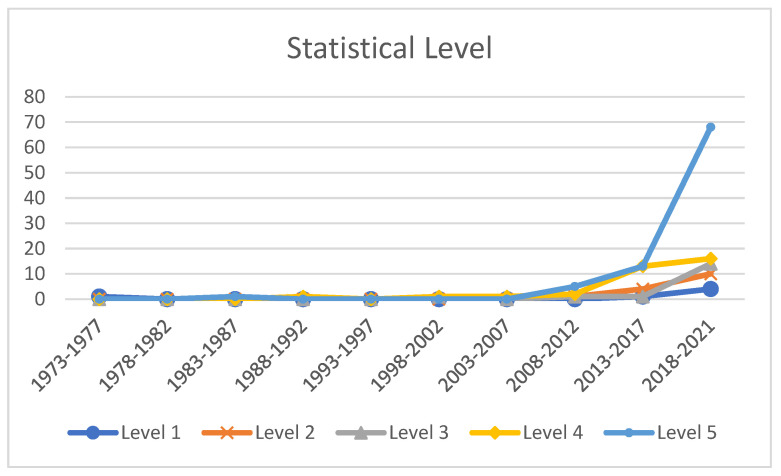
Trend of Statistical Level by Five-Year Interval during 1973–2021 (n = 159).

**Table 1 ijerph-19-12342-t001:** Trend of Data Location by Five-Year Interval during 1973–2021.

	Mainland China	Hong Kong	India	Israel	Multiple Countries	Turkey	Malaysia	Iran	Pakistan	Philippines
1973–1977	0	0	1	0	0	0	0	0	0	0
1978–1982	0	0	0	0	0	0	0	0	0	0
1983–1987	0	0	0	1	0	0	0	0	0	0
1988–1992	0	0	0	1	0	0	0	0	0	0
1993–1997	0	0	0	0	0	0	0	0	0	0
1998–2002	0	0	0	0	1	0	0	0	1	0
2003–2007	0	0	0	1	0	0	0	0	0	0
2008–2012	1	4	0	1	1	1	0	0	0	0
2013–2017	1	10	3	4	1	4	3	2	3	1
2018–2021	34	11	14	9	13	9	9	7	3	6
Total	36	25	18	17	16	14	12	9	7	7

*Note.* n = 167.

**Table 2 ijerph-19-12342-t002:** Volume of Publications by Themes during 1973–2021.

Nature	No. & %	Antecedent	No. & %	Consequence	No. & %
Theory	35, 20.83%	Contextual	5, 2.98%	Teacher Personal Outcome	32, 19.05%
Definition	35, 20.83%	School-level	68, 40.48%	Teaching Practice	6, 3.57%
Content	23, 13.69%	Personal	106, 63.10%	Student Outcome	4, 2.38%
Model	3, 1.79%	-	-	School Level Outcome	2, 1.19%
Measures	5, 2.98%	-	-	-	-
Total	68, 40.48%	-	140. 83.33%	-	41, 24.4%

*Note.* n = 168.

**Table 3 ijerph-19-12342-t003:** Modified Descriptions of Five Statistical Levels from Hallinger (2011).

Level	Description
1	Descriptive. The use of numbers to represent central tendencies and/or variability of scores.
2	Single causal factor–correlational. The examination of the relationship or association between two variables, one of which presumably co-varies with or influences the other.
3	Single causal factor–correlational with controls. The examination of the relationship between two variables while controlling for the influence of one or more other variables.
4	Multiple factors. This involves probing the differential effects of multiple sources of influence on a particular variable.
5	Advanced modelling. This comprises tests that explore relationships among multiple independent and dependent variables in a manner that allows for the examination of moderating and/or mediating effects.

**Table 4 ijerph-19-12342-t004:** Trend of Frequency of Top 6 Qualitative Data Collection Techniques by Five-Year Interval during 1973–2021.

	Interview	Qualitative Survey	Classroom Observation	Case Study	Reflective Reports	Journal Writing
1973–1977	0	0	0	0	0	0
1978–1982	0	0	0	0	0	0
1983–1987	0	0	0	0	0	0
1988–1992	0	0	0	0	0	0
1993–1997	0	0	0	0	0	0
1998–2002	0	0	0	0	0	0
2003–2007	1	0	0	0	0	0
2008–2012	0	0	0	0	0	0
2013–2017	4	0	1	0	0	0
2018–2021	11	2	1	1	1	1
Total	16	2	2	1	1	1

*Note.* n = 20.

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
