# Peer review of "Trajectory of Teacher Well-Being Research between 1973 and 2021: Review Evidence from 49 Years in Asia"

_ijerph, 2022, doi:10.3390/ijerph191912342_

Round 1
Reviewer 1 Report
The manuscript focuses on analyzing the scientific literature published in the last 49 years across 46 countries and regions in Asia on the teacher well-being topic with the aim to provide recommendations for future research that could help narrowing the knowledge gap in a research area which currently draws significant interest among scholars and policymakers. The authors made a notable attempt in providing a thorough research on the trend in the teacher well-being literature in Asia and seems to reach some interesting and promising insights.
Meanwhile, the following issues needs to be addressed:
1. First inclusion criterium “journal articles needed to be published between 1973 and 2021 in Asia” is not very clear. What does it mean published in Asia? The journals, the authors, the data research sites, or the participants should be located in Asia? The authors should clear specify if published in Asia is about data location or something else.
2. It is not very clear how exactly 10,466 articles were identified. There were 168 articles in the initial corpus (based on Scopus and Google Scholar searching) and then this corpus suudenly increased to 10466 articles. All these additional articles are retrieved from other sources or databases? It is not very clear which are these other sources.
3. Lines 276-277: “The residual research production was contributed by the remaining 36 regions around the world”. Shouldn’t be around Asia? In this case, the two articles referring to Argentina should be excluded from research.
4. For consistency reasons, the title of the 5th column in Table 2 should be consequences as stated in text (line 298) and not effect.
5. Since 167 out of 168 articles are empirical, I don’t think that the analysis based on research type (chapter 4.4) is relevant.
6. Figure 6: maybe more distinct colors should be used to code the 5 levels of statistical methods (the different shades of grey are not very easy to distinct among each other)
7. There is an issue with the structure of the paper. I assume that chapters 6,7,8,9, 10 are in fact sub-chapters of chapter 5 Discussion and future research. In this case they should be renumbered as 5.1, 5.2, 5.3, 5.4 and 5.5
8. There are some sentences in the manuscript are not very clear and concise. (e.g lines 56-57, 256-257). There are some grammatical mistakes. (e.g lines 106, 169-170)
Author Response
We appreciated Reviewer 1's encouraging comments. Please see the responses attached. Best wishes.

Reviewer 2 Report
The manuscript addresses a very interesting international topic that can help to improve teacher performance, under the title " Trajectory of teacher well-being research between 1973 and 2 2021: review evidence from 49 years in Asia ". The topic of the manuscript, as well as the updated literature review provided by the authors, is appropriate.
The overall assessment of the paper is positive. The authors address the topic of the study in a clear way and rely on a review of the scientific literature that is comprehensive and current. The paper is well-structured and easy to follow.
Overall, it is a good piece of work. Thus, I recommend the manuscript for publication.
Author Response

(The authors gave the same response as above.)

Reviewer 3 Report
Teacher well-being is an important topic to investigate. The idea of a well-being trajectory keeps the focus on long term needs of the profession in a complex fast changing society, and a review of methodological approaches may uncover gaps in research approaches that could generate new insight.
The findings clearly show the recent increased interest in teacher well-being research.
At the moment, while it has merit, the paper is too long, wordy, and difficult to read.
Edit for clarity and conciseness. There are too many redundant words and examples of incorrect word use or phrasing. This negatively impacts the overall quality of the paper and makes it hard for the reader to follow the arguments. For example:
In line 11 did the database identify the articles? More correct to say ‘A search of the Scopus database identified….’
Fix the run-on sentence in line 31
Lines 37-38 do not make sense
Lines 39-41 - retention is not the best word to choose here. Fix the run-on sentence.
line 43 more correct to say ‘recently’ or ‘in the past two years.’
line 44 better to say ‘However, the main contributors in this field have been’
line 45-48 - this sentence needs to be rewritten to improve clarity.
I am making no more suggestions after page 1 but this level of close editing is needed throughout the paper. ‘Pulverised comprehension’ in line 68-69 is another example of incorrect word use…
Here is another example of how to edit for clarity and conciseness in lines 106 and 107:
“There are the five aspects that outline the relevance of investigating TWB in the Asian 106 context. The first relevance is that the construct of TWB is context-dependent.”
Change this to:
“Five aspects are of relevance in investigating TWB in the Asian context. The first is that….”
Another example of unnecessary wordiness is from line 620: “Firstly, a limitation that needs to be discussed is the interpretation…” This could just be “Firstly, the interpretation…”
I recommend that the authors get support or use a tool to improve academic writing and academic English. The Manchester Academic Phrasebank is freely available online and is one such tool.
The authors could make use of bullet points to summarise information and reduce wordiness. An example is section 3.1.
Another suggestion for reducing wordiness in section 4.3 is to include the paper examples of the themes in table 2. The paragraph is dense and hard to read. Seek at all times in all sections not to repeat information that is or could be included in tables.
I recommend checking carefully for repetition - results seem to be summarised again and so repeated in the discussion section.
In the conclusion - I feel more could be done here to comment on why some methodologies may be more useful for understanding some aspects of teacher well-being than others.
Author Response
We appreciated Reviewer 3's encouraging comments. Please see the responses attached. Best wishes.

Round 2
Reviewer 3 Report
The authors have attended to the suggestions made.